# Air pollution exposure associates with increased risk of neonatal jaundice

Liqiang Zhang [1,8], Weiwei Liu [2,8], Kun Hou[1,8], Jintai Lin [3], Changqing Song[1], Chenghu Zhou[4], Bo Huang [5], Xiaohua Tong[6], Jinfeng Wang [4], William Rhine [7], Ying Jiao[2], Ziwei Wang [3], Ruijing Ni[3], Mengyao Liu[3], Liang Zhang[1], Ziye Wang[1], Yuebin Wang [1], Xingang Li[1], Suhong Liu[1] & Yanhong Wang[2]

Clinical experience suggests increased incidences of neonatal jaundice when air quality worsens, yet no studies have quantified this relationship. Here we reports investigations in 25,782 newborns showing an increase in newborn's bilirubin levels, the indicator of neonatal jaundice risk, by 0.076 (95% CI: 0.027–0.125), 0.029 (0.014–0.044) and 0.009 (95% CI: 0.002–0.016) mg/dL per $\mu g/m^3$ for $PM_{2.5}$ exposure in the concentration ranges of 10–35, 35–75 and 75–200 $\mu g/m^3$, respectively. The response is 0.094 (0.077–0.111) and 0.161 (0.07–0.252) mg/dL per $\mu g/m^3$ for $SO_2$ exposure at 10–15 and above 15 $\mu g/m^3$, respectively, and 0.351 (0.314–0.388) mg/dL per $mg/m^3$ for CO exposure. Bilirubin levels increase linearly with exposure time between 0 and 48 h. Positive relationship between maternal exposure and newborn bilirubin level is also quantitated. The jaundice—pollution relationship is not affected by top-of-atmosphere incident solar irradiance and atmospheric visibility. Improving air quality may therefore be key to lowering the neonatal jaundice risk.

[1] State Key Laboratory of Earth Surface Processes and Resource Ecology, Beijing Normal University, 100875 Beijing, China. [2] Beijing Obstetrics and Gynecology Hospital, Capital Medical University, 100006 Beijing, China. [3] Laboratory for Climate and Ocean-Atmosphere Studies, Department of Atmospheric and Oceanic Sciences, School of Physics, Peking University, 100871 Beijing, China. [4] State Key Laboratory of Resources and Environment Information System, Institute of Geographical Science and Natural Resources, Chinese Academy of Sciences, 100101 Beijing, China. [5] Department of Geography and Resource Management, The Chinese University of Hong Kong, 999077 Hong Kong, China. [6] School of Surveying and Geo-informatics, Tongji University, 200092 Shanghai, China. [7] Division of Neonatology, Lucile Packard Children's Hospital Stanford University School of Medicine, Stanford, CA 94304, USA. [8] These authors contributed equally: Liqiang Zhang, Weiwei Liu, Kun Hou. Correspondence and requests for materials should be addressed to L.Z. (email: zhanglq@bnu.edu.cn) or to J.L. (email: linjt@pku.edu.cn) or to C.S. (email: songcq@bnu.edu.cn) or to C.Z. (email: zhouch@lreis.ac.cn)

Air pollution is a serious problem in mainland China. In the East China-wide haze events in January 2013, the highest hourly concentration of ambient fine particulate matter (PM$_{2.5}$) exceeded 1000 μg per m$^3$ in Beijing[1]. Exposure to air pollution is linked to various respiratory diseases[2,3], chronic obstructive pulmonary disease[4,5], asthma[6], lung cancer[7,8], and increase in death risk[9,10]. Pregnant women, developing fetuses and newborns are especially susceptible and vulnerable to environmental pollution[11,12]. There is increasing evidence of harmful effects of air pollution exposure on newborns[13–17], such as low birth weight[18], elevated systolic blood pressure[19] and mortality[20].

Neonatal jaundice is the most common clinical problem of newborns. Severe neonatal jaundice and its progression to acute bilirubin encephalopathy and kernicterus become the leading cause of newborn re-hospitalization, cerebral palsy, pathogenesis of deafness and bradylalia[15,21–23]. The leading health-care policy research groups like the Child Health Epidemiology Reference Group of the World Health Organization (WHO) and the Global Burden of Disease Collaborators have increasingly recognized the clinical and public health significance of neonatal jaundice as an important neonatal condition that deserves global health attention in the post-2015 millennium development goals era[24–26]. Known risk factors of neonatal jaundice include intrauterine retardation[27], gestational diabetes[28], sepsis[27], intrauterine infections[29], pregnancy anemia[27], and congenital hypothyroidism[30]. It is known that exposure of pregnant women to environmental tobacco smoke is associated with the risk of neonatal jaundice[31–33]. However, the correlation between air quality and the neonatal jaundice risk remains unquantified. This study attempts to assess the potential impacts of air pollution exposure on the risk of neonatal jaundice as well as the magnitude and mechanisms of these impacts.

We collected maternal and neonatal clinical data from Beijing, China. A total of 25,782 term singleton newborns without hemolytic disease and less than 7 days of age from June 2014 through May 2017 were examined. Beijing has diverse terrains[34] and a large range of air quality conditions across space and time[35,36]. Although the average (Supplementary Fig. 1a) and maximum (Supplementary Fig. 1b) air pollution level is quite high (exceeding 100 μg per m$^3$ at many places), the minimum pollution level is below 6 μg per m$^3$ (Supplementary Figure 1c). This large range of pollution severity (also see Supplementary Fig. 2) in Beijing provides an excellent opportunity to study the association between air pollution exposure and neonatal jaundice incidence. This study indicates air pollution exposure is associated with increased risk of neonatal jaundice, and improving air quality may be key to lowering the neonatal jaundice risk.

## Results

**Linking air pollution exposure to the jaundice risk**. According to the air quality standard of China, levels of air quality are classified into excellent, good, slightly polluted, moderately polluted, heavily polluted and severely polluted[37]. Air quality index (AQI) was issued by the MEE to designate the overall air quality, by considering concentrations of multiple pollutants. The correlations between different air pollutant concentrations were presented in Supplementary Table 2. The AQI is below 100 at Levels 1 and 2, which is designated here as having relatively good air quality, although the actual pollutant concentrations at Levels 1 and 2 may still be high according to the WHO guidance[38]. The AQI exceeds 100 for Levels 3–6, which is designated as having relatively bad air quality.

We examined the relationship between air pollution exposure and the neonatal jaundice risk. Bilirubin levels of the newborns

were measured by neonatologists using the transcutaneous bilirubin (TCB) meters (Type: JH20–1C). The meters were calibrated rigorously prior to use. For the jaundiced newborns requiring treatment, their bilirubin levels were measured before they receive phototherapy. The dataset with 25,782 newborns was divided into two groups with jaundice ($n = 14,058$) and without jaundice ($n = 11,724$) according to the Chinese clinical guideline of neonatal jaundice[33] as shown in Supplementary Table 1. Supplementary Table 3 summarizes sociodemographic and health characteristics of the newborns in the two groups. Statistically significant differences exist between the two groups ($P < 0.05$) in the occurrence of relatively bad air (daily AQI > 100) during the observation period. This suggests potential associations between these factors and incidence of neonatal jaundice.

**Links between individual pollutants and the jaundice risk**. We divided the 14,058 jaundiced newborns into two subgroups according to the Chinese clinical guideline of neonatal jaundice[33]. As shown in Supplementary Table 1, the first subgroup (Subgroup I) experienced physiological neonatal jaundice, which usually disappears without treatments. The second subgroup (Subgroup II) required close monitoring and prompt treatments. For Subgroup II, Supplementary Table 4 listed the jaundiced newborns who were considered to take phototherapy (those who were considered for phototherapy usually received phototherapy. If they did not receive phototherapy, their bilirubin levels were to be closely monitored. Once their bilirubin levels reached the degree of the required phototherapy, they had to receive phototherapy) and the newborns who were required to take phototherapy (and actually took it). We then assessed the relationship between jaundice severity and individual pollutants.

Supplementary Table 5 shows that concentrations of individual air pollutants (average and maximum values from the day of birth to the day before the peak bilirubin level was measured) for the close monitoring subgroup (Subgroup II) were higher than those in Subgroup I. The inter-group difference in mean concentration was statistically significant for PM$_{2.5}$ and sulfur dioxide (SO$_2$); and the inter-group difference in maximum concentration was statistically significant for PM$_{2.5}$, SO$_2$ and carbon monoxide (CO). These results further supported that more severe jaundice tended to occur in more polluted air environments.

We utilized the generalized additional model (GAM) to quantitatively link the pollutant concentrations (average of daily mean pollution from the day of birth to the day before the peak bilirubin level was measured) and the peak bilirubin levels of the newborns. The $R^2$ statistic was utilized to measure the explanatory power of the final GAM. Figure 1 illustrates the associations of PM$_{2.5}$, SO$_2$ and CO concentrations with the peak bilirubin levels of the newborns, respectively. Individual relationships between individual air pollutants and the peak bilirubin levels were assessed using the partial response plots (PRPs) and marginal effects. The marginal effect refers to $\exp(s(x))-1$, where $x$ is the air pollutant variable of interest, and $s(x)$ is the corresponding smooth function. The partial residuals plot reflected the effects of each air pollutant on bilirubin levels. The results show that the associations between pollutant concentration and the peak bilirubin level were different in different pollutant concentration intervals.

We examined the link between PM$_{2.5}$ exposure and the peak bilirubin level of each newborn. Based on the WHO air quality guidelines and Chinese air quality standards, we divided PM$_{2.5}$ concentrations into five intervals: 0–10, 10–35, 35–75, 75–200, and >200 μg per m$^3$. Figure 1a shows the relationship between PM$_{2.5}$ concentrations (average of daily pollution from the day of birth to the day before the peak bilirubin level was measured) and

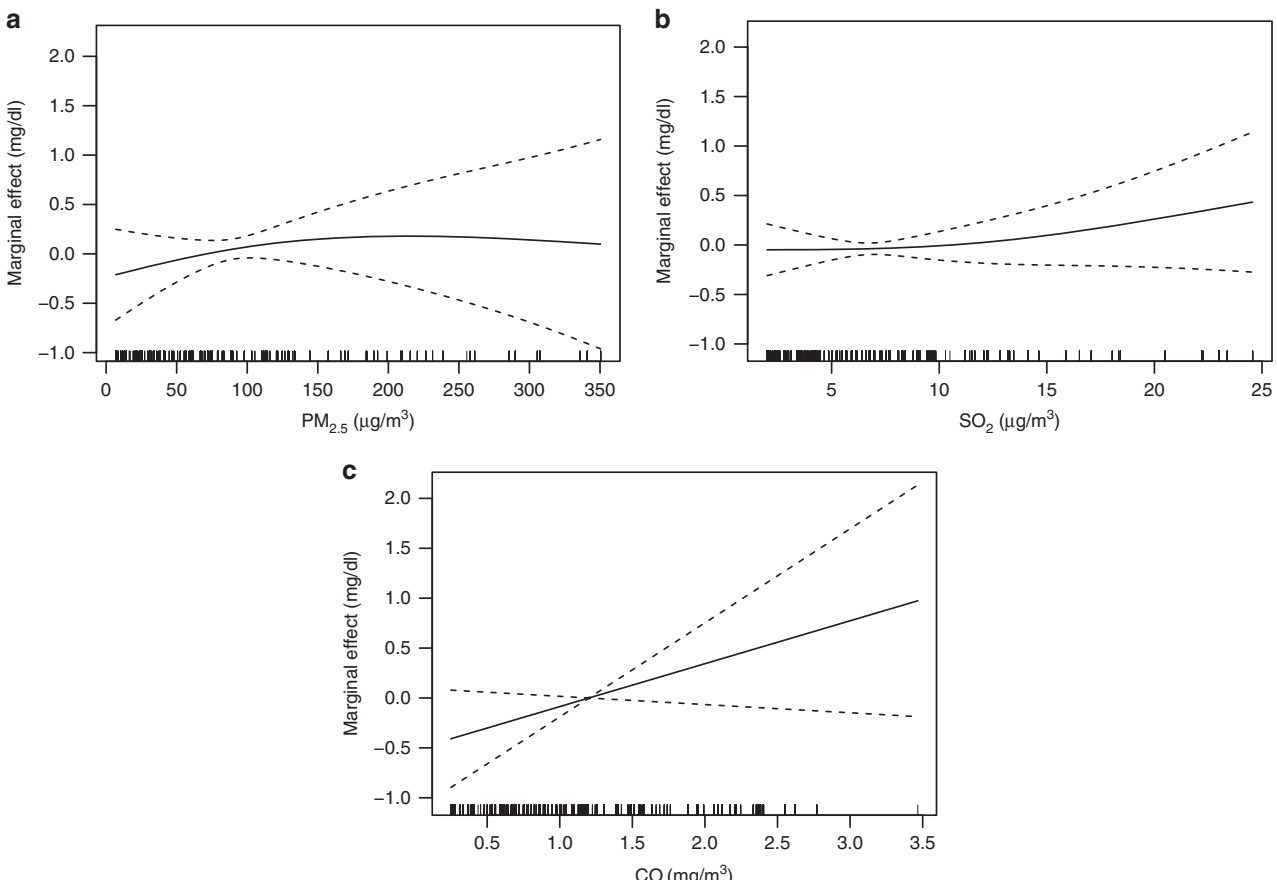

**Fig. 1** Partial response plots for the peak bilirubin level with respect to average concentration of PM$_{2.5}$, SO$_2$ and CO. The *y*-axis represents the marginal effects. The *x*-axis represents average concentrations of PM$_{2.5}$ (**a**), SO$_2$ (**b**) and CO (**c**). The dashed lines represent 95% confidence interval. The vertical lines adjacent to the *x*-axis represent the frequency of the data

| Table 1 Association of PM$_{2.5}$ exposure with the peak bilirubin levels on the basis of an increase of 1.0 µg per m$^3$ (95 % CI) in exposure to PM$_{2.5}$ | | | | |
|---|---|---|---|---|
| **Exposure intervals (µg per m$^3$)** | **Risk in peak bilirubin levels (mg per dL)** | **Confidence lower limit (mg per dL)** | **Confidence upper limit (mg per dL)** | ***P* value** |
| (0, 10) | 0.848 | −0.574 | 2.269 | 0.157 |
| (10, 35) | 0.076 | 0.027 | 0.125 | 0.003 |
| (35,75) | 0.029 | 0.014 | 0.044 | 0.031 |
| (75, 200) | 0.009 | 0.002 | 0.016 | 0.008 |
| >200 | 0.01 | −0.008 | 0.028 | 0.435 |

For PM$_{2.5}$ concentrations ∈ [10, 35] µg per m$^3$, a 1.0 µg per m$^3$ increase in PM$_{2.5}$ concentrations was associated with a 0.076 mg per dL (95% CI: 0.027–0.125) rise in the peak bilirubin level. For a 1.0 µg per m$^3$ increase in PM$_{2.5}$ concentration, the peak bilirubin level increased by 0.029 mg per dL (95% CI: 0.014–0.044) for PM$_{2.5}$ concentrations ∈ (35, 75] µg per m$^3$, and by 0.009 mg per dL (95% CI: 0.002–0.016) for PM$_{2.5}$ concentrations ∈ (75, 200] µg per m$^3$. The relationship between PM$_{2.5}$ concentrations and the neonatal jaundice risk nearly plateaued at concentrations exceeding 200 µg per m$^3$, such that an additional increase in pollution concentration was not statistically significantly associated with a further increase in bilirubin level

the peak bilirubin levels of the newborns (the dashed lines represent 95% CI in Fig. 1). A spline analysis for Fig. 1a suggested that the PM$_{2.5}$–bilirubin level relationship was not statistically significant at concentrations below 10 µg per m$^3$, then increased with increasing PM$_{2.5}$ concentrations until 200 µg per m$^3$, and finally nearly plateaued and was statistically insignificant as PM$_{2.5}$ concentration exceeded 200 µg per m$^3$.

The results in Table 1 show that for PM$_{2.5}$ concentrations ∈ (10, 35] µg per m$^3$, a 1.0 µg per m$^3$ increase in PM$_{2.5}$ concentrations (average of daily pollution from the day of birth to the day before the peak bilirubin level was measured) was associated with a 0.076 mg per dL (95% CI: 0.027–0.125) rise in the peak bilirubin level. For a 1.0 µg per m$^3$ increase in PM$_{2.5}$ concentration, the peak

bilirubin level increased by 0.029 mg per dL (95% CI: 0.014–0.044) for PM$_{2.5}$ concentrations ∈ (35, 75] µg per m$^3$, and by 0.009 mg per dL (95% CI: 0.002–0.016) for PM$_{2.5}$ concentrations ∈ (75, 200] µg per m$^3$. The relationship between PM$_{2.5}$ concentrations and the neonatal jaundice risk nearly plateaued at concentrations exceeding 200 µg per m$^3$, such that an additional increase in pollution concentration was not statistically significantly associated with a further increase in bilirubin level.

Table 2 presents the association between SO$_2$ concentrations (average of daily pollution from the day of birth to the day before the peak bilirubin level was measured) and the peak bilirubin levels. For SO$_2$ concentrations below 10 µg per m$^3$, the SO$_2$–bilirubin level association was not statistically significant,

**Table 2 Association of SO$_2$ with bilirubin levels for a 1.0 μg per m$^3$ increase**

| Exposure intervals (μg per m$^3$) | Estimated risk in peak bilirubin levels (mg per dL) | Confidence lower limit (mg per dL) | Confidence upper limit (mg per dL) | P value |
|---|---|---|---|---|
| (0, 5) | 0.082 | −0.157 | 0.321 | 0.327 |
| (5, 10) | 0.028 | −0.113 | 0.17 | 0.776 |
| (10, 15) | 0.094 | 0.077 | 0.111 | <0.001 |
| >15 | 0.161 | 0.07 | 0.252 | <0.001 |

For SO$_2$ concentrations below 10 μg per m$^3$, the SO$_2$–bilirubin level association was not statistically significant. For SO$_2$ concentrations ∈ (10, 15) (μg per m$^3$), a 1.0 μg per m$^3$ increase in SO$_2$ concentration was associated with a 0.094 mg per dL (95% CI: 0.077–0.111) rise in the peak bilirubin level. And for SO$_2$ concentrations above 15 μg per m$^3$, a 1.0 μg per m$^3$ increase in SO$_2$ concentration was associated with a 0.161 mg per dL (95% CI: 0.07–0.252) rise in the peak bilirubin level

consistent with the result in Fig. 1b. For SO$_2$ concentrations ∈ (10, 15) (μg per m$^3$), a 1.0 μg per m$^3$ increase in SO$_2$ concentration was associated with a 0.094 mg per dL (95% CI: 0.077–0.111) rise in the peak bilirubin level, and for SO$_2$ concentrations above 15 μg per m$^3$, a 1.0 μg per m$^3$ increase in SO$_2$ concentration was associated with a 0.161 mg per dL (95% CI: 0.07–0.252) rise in the peak bilirubin level.

As shown from Fig. 1c, CO concentrations had a linear relationship with neonatal bilirubin levels. Considering the small variation of CO concentrations, the analysis was conducted for CO range of (0, 3.5) (mg per m$^3$). The peak bilirubin level increased 0.351 mg per dL (95% CI: 0.314–0.388) as CO concentration (average of daily pollution from the day of birth to the day before the peak bilirubin level was measured) increased by 1.0 mg per m$^3$.

**Associating neonatal exposure time and jaundice risk**. The course of neonatal jaundice is short, and physiological jaundice peaks at 4–6 days after birth. Continuous exposure to air pollution may have a high negative influence on the risk of neonatal jaundice. To validate this, we investigated the relationship between continuous air pollution exposure and the risk of neonatal jaundice. For each of PM$_{2.5}$, SO$_2$ and CO, we assessed the association between the time of neonatal exposure to air pollution and the risk of neonatal jaundice; we calculated the exposure time (from the hour of birth to the hour when the peak bilirubin level was measured) for each newborn.

After the influence on the peak bilirubin levels of average pollutant concentration and the interaction with the exposure time was controlled (see Methods), the relationship between the exposure time and the peak bilirubin level of each newborn was determined. The curve analysis in Fig. 2 suggests that similarly for PM$_{2.5}$, SO$_2$ and CO, the risk of neonatal jaundice increased linearly with the exposure time from 0 to 48 h (the dashed lines represent 95% CI in Fig. 2). The increased rate of the peak bilirubin level gradually slowed down from 48 to 120 h of exposure, and then nearly plateaus as exposure time exceeded 120 h (5 days).

Previous studies suggested that bilirubin levels peak in neonates at about 96 h of life, which was typically after newborns were discharged from hospitals[39,40]. Our finding provides partial explanations based on air pollution exposure time. Health-care professionals should regularly monitor neonatal bilirubin levels within the subsequent 48−120 h of age[22,41], especially in air pollution conditions.

**Associating maternal exposure and neonatal jaundice risk**. Maternal exposure to air pollution during pregnancy might inhibit newborn's bilirubin metabolism. On the one hand, the fetus is highly sensitive to maternal exposure to air pollution at pregnancy, due to the susceptibility of target organs and systems during developmental periods of life[3,42]. On the other hand, maternal exposure to air pollution can cause placental

inflammation[43,44] and thus enhance the risk of neonatal jaundice. To investigate the possible adverse impacts of maternal exposure to air pollution during pregnancy on newborn's metabolism of bilirubin, we assessed the correlation between maternal exposure to air pollution during the third trimester of pregnancy and the risk of jaundice in newborns (see Methods). Table 3 shows that maternal exposure to each of PM$_{2.5}$, SO$_2$ and CO had statistically significant, positive correlation with the severity of neonatal jaundice.

We used the GAM to quantitate the relationship between maternal exposure to air pollution during the third trimester of pregnancy and the peak bilirubin level of each newborn. Figure 3a shows that the increase of maternal exposure to PM$_{2.5}$ was associated with the increased peak bilirubin levels (the dashed lines represent 95% CI in Fig. 3). The peak bilirubin levels increased slowly with the rising PM$_{2.5}$ concentrations from 0 to 70 μg per m$^3$ and rapidly afterwards. Figure 3b, c also shows positive relationships between maternal exposure to SO$_2$ (Fig. 3b) and CO (Fig. 3c) and the peak bilirubin levels. Because infection was associated with neonatal jaundice, higher in utero air pollutant (PM$_{2.5}$, SO$_2$ and CO) exposure may be associated with higher neonatal jaundice through increasing the risk of infection.

**Sensitivity analysis**. In the above sections, we assessed the influences of the duration of neonatal exposure to air pollution, air pollutant concentration newborns were exposed to, and maternal exposure to air pollution in the third trimester of pregnancy on the risk of neonatal jaundice, respectively. Here we estimated the sole effect of each factor on neonatal jaundice after the other two factors were controlled (see Methods).

We find that the relationship between each factor and neonatal jaundice, after controlling the other two factors, were very similar to the relationship without controlling the other two factors (Supplementary Figs. 3–5), suggesting that the influences of individual factors on neonatal jaundice were largely independent.

In this study population, the newborns stayed in the wards all the time before they were discharged from the hospital, and thus they were not exposed to outdoor sunlight. As a result, we found that the top-of-atmosphere (TOA) incident solar irradiance did not affect the relationship between air pollution exposure and the incidence of neonatal jaundice, by stratified analysis of TOA irradiance (see Methods, Supplementary Table 8).

Supplementary Table 6 shows that in a polluted environment (AQI > 100), atmospheric visibility was much lower than that under less polluted environments (AQI < 100). Supplementary Fig. 6 describes a clear power law relationship between daily mean PM$_{2.5}$ concentration and daily mean visibility (Visibility = $165.34 \times PM_{2.5}^{-0.73}$). A spline analysis shows that the extent of visibility reduction for a unit of PM$_{2.5}$ concentration enhancement becomes insignificant as PM$_{2.5}$ exceeds 200 μg per m$^3$. We find the association between air pollution exposure and jaundice is similar at different levels of visibility, as shown in the stratified analysis of visibility (see Methods, Supplementary Table 9).

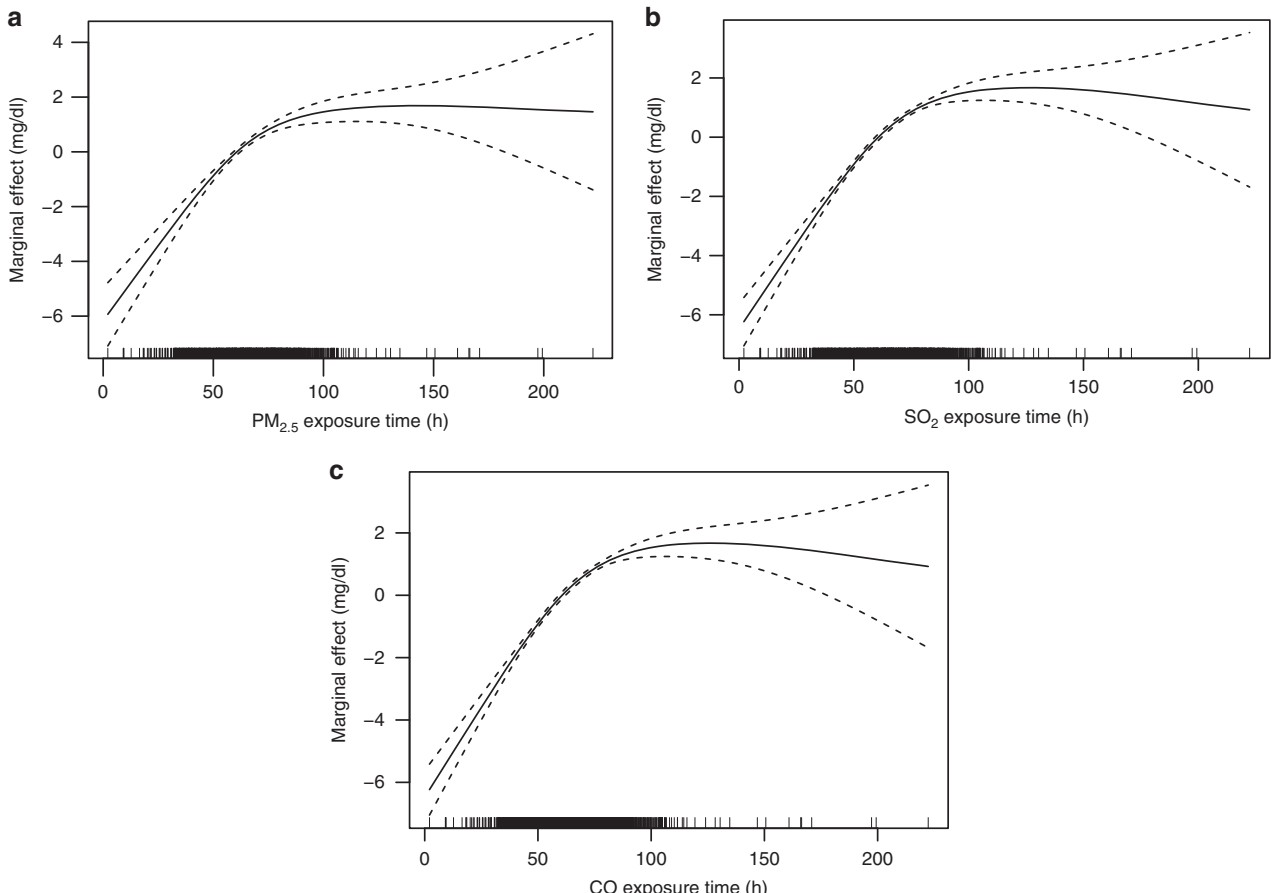

**Fig. 2** Partial response plots for the peak bilirubin levels with respect to pollution exposure times of PM$_{2.5}$, SO$_2$ and CO. The *y*-axis represents the marginal effects. The dashed lines represent 95% CI. The *x*-axis represents PM$_{2.5}$ exposure time in (**a**), SO$_2$ exposure time in (**b**), and CO exposure time in (**c**). The vertical lines adjacent to the *x*-axis represent the frequency of the data

**Table 3 The correlation between maternal exposure to air pollution during the third trimester of pregnancy and the risk of jaundice in newborns**

|  | (Subgroup I) (*n* = 7722) | (Subgroup II) (*n* = 6336) | *T* value | *P* value |
|---|---|---|---|---|
| PM$_{2.5}$ (µg per m$^3$) | 69.98 ± 16.06 | 73.79 ± 16.76 | −13.455 | 0.000 |
| SO$_2$ (µg per m$^3$) | 4.71 ± 2.19 | 5.41 ± 2.47 | −17.564 | 0.000 |
| CO (mg per m$^3$) | 1.22 ± 4.06 | 1.77 ± 7.41 | −5.513 | 0.000 |

Maternal exposure to each of PM$_{2.5}$, SO$_2$ and CO had statistically significant, positive correlation with the severity of neonatal jaundice

In addition to the presented exposure intervals, we added an overall linear model (Supplementary Eq. 1) for the entire exposure range. Then, we analyzed the data and presented the results as shown in Supplementary Tables 10–11. The TOA irradiance had a weak negative correlation with bilirubin levels (see Supplementary Tables 10). As the value of the TOA irradiance enhanced, bilirubin levels decreased very slightly. Consistent with the results in Supplementary Table 8, the TOA irradiance had little impact on the incidence of neonatal jaundice. As shown in Supplementary Table 11, atmospheric visibility was not statistically significantly associated with the increase in bilirubin levels ($P > 0.05$). Thus, it indicates that bilirubin levels did not change with different levels of atmospheric visibility, consistent with the results listed in Supplementary Table 9. From the above analysis, we conclude that our findings about effects of the TOA irradiance and atmospheric visibility on the incidence of neonatal jaundice were robust and plausible.

## Discussion

The above analysis suggests that neonatal exposure to air pollution might significantly increase the neonatal jaundice risk. Possible causal mechanisms are presented below.

Sunlight is effective in breaking down bilirubin levels of newborns[45–47]. However, newborns stayed in the wards and were not exposed to outdoor sunlight, as confirmed by examining the relationship between air pollution exposure and jaundice by stratified analysis of TOA solar irradiance (see Methods, Supplementary Table 8).

The biological mechanism underpinning the association concerns how breathed air pollutants affect jaundice through multiple routes. There is scientific evidence that air pollutants enter the bloodstream where they possibly interact with the organs including liver tissue to produce pathological effects[48]. Breathed PM$_{2.5}$ might change the levels and activities of P450 enzyme of human cytochrome, which plays an important role in the

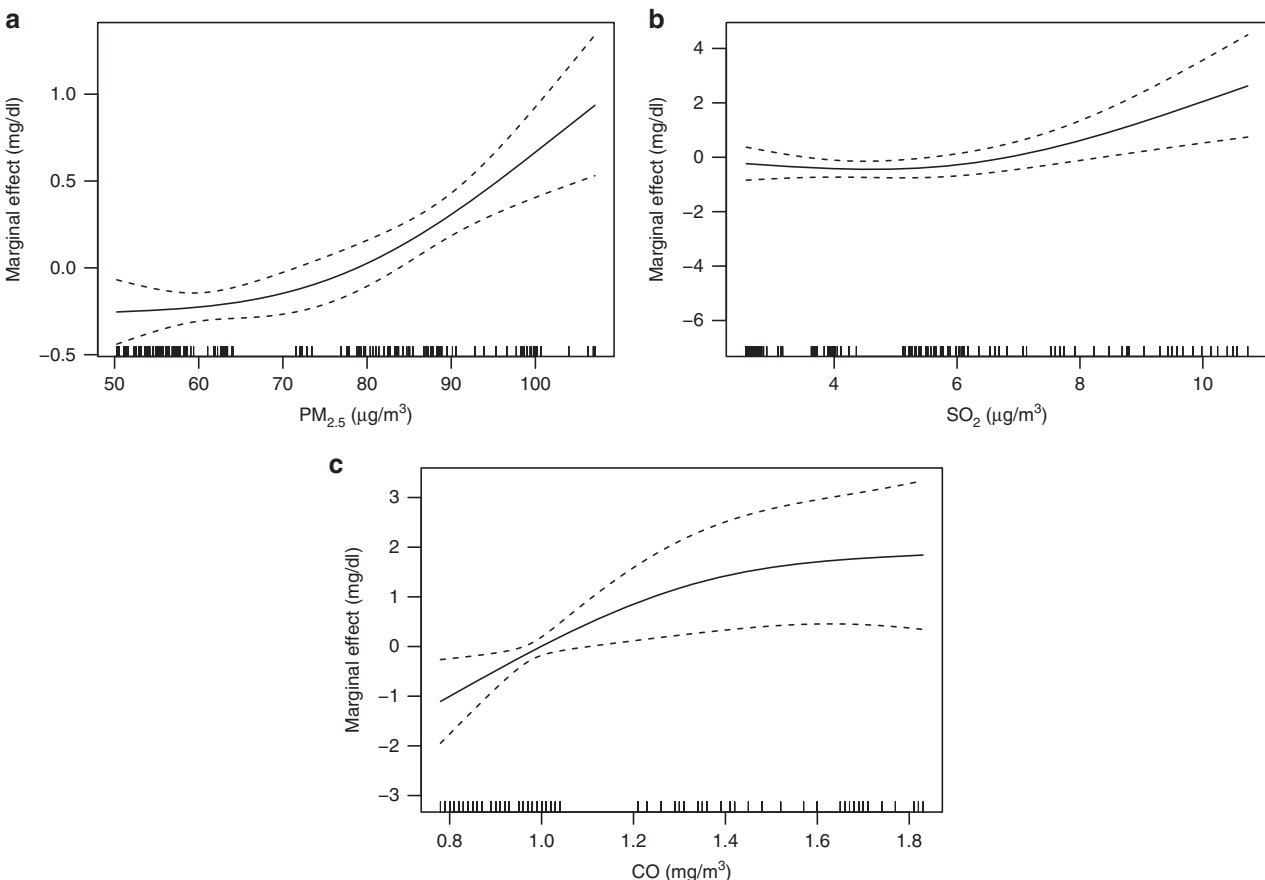

**Fig. 3** Partial response plots for the peak bilirubin level with maternal exposed to PM$_{2.5}$, SO$_2$ and CO. The *y*-axis represents the marginal effects. The *x*-axis represents average concentrations of PM$_{2.5}$ (**a**), SO$_2$ (**b**) and CO (**c**). The dashed lines represent 95% CI. The vertical lines adjacent to the *x*-axis represent the frequency of the data

bilirubin metabolism in newborns[49]. Through affecting the P450 enzyme, breathed PM$_{2.5}$ can cause serum bilirubin aggregation and thus raise the bilirubin levels of newborns. Also, air pollutants may affect the metabolism of the bilirubin through damaging the liver function. Studies in experimental models and humans presented the accumulation in the blood and liver following pulmonary exposure to a broader size range of nanoparticles, with translocation markedly greater when sufficiently small, such as for particles <10 nm diameter[48,50–53]. PM$_{2.5}$ has direct adverse effects on the liver function[54,55]. SO$_2$ is a systemic oxidative damage agent, and it may cause toxicological damage to multiple organs like brain, lung, heart and liver of animals[56]. CO might hinder metabolic and transport function of the placenta[57] and concentrate more in the fetus than in the mother[58] after crossing the placental barrier. Moreover, CO also leads to acute hepatic dysfunction[59] and thus affects the function.

To validate the liver-associated route, we examined the levels of alanine aminotransferase (ALT), γ-Glutamyl transferase (GGT) and aspartate aminotransferase (AST) of 300 newborns (independent of the 25,782 newborns). ALT, AST and GGT can cause abnormal liver function. These 300 newborns had very high total serum bilirubin (TSB) levels in blood (as gold standard for jaundice diagnosis), and they were hospitalized in the neonatal intensive care unit. Among these severely jaundiced newborns, 129 were exposed to more polluted air (AQI > 100), and the other 171 were exposed to less polluted air (AQI < 100). Supplementary Table 7 shows that the ALT and GGT levels were higher in newborns exposed to more serious air pollution (AQI > 100), although the difference was not statistically significant likely due

to small sample size. Moreover, newborns exposed to poor air quality had a much higher AST level than those breathing cleaner air (88.68 ± 67.48 versus 62.85 ± 33.56 U per L, *P* = 0.014). The scatter plot in Supplementary Fig. 7 further shows that among the 300 newborns, the peak bilirubin level grew as the AQI (including all pollutants) increased. These results suggested a significant association between air pollution exposure, neonatal liver functions, and neonatal jaundice.

This study estimated the impact of neonatal and maternal air pollution exposure on the neonatal jaundice risk, and provided evidence for the association to resolve concerns about causal inference. We synthesized the available evidence to quantitate the neonatal jaundice risk associated with PM$_{2.5}$, SO$_2$ and CO including exposure time and average concentration. Our analyses control individual-level differences in maternal pregnancy-induced outcomes like gestational diabetes mellitus, air temperature and relative humidity as confounding factors, and lessen the concern about the confounding. While newborns were in the ward and therefore had little exposure to outdoor sunlight, mothers were exposed to different levels of sunlight during pregnancy, which could confound the air pollution effects for maternal exposure. In addition to environmental factors, mother's socioeconomic and behavioral characteristics may play significant roles in neonatal jaundice risk. In future work, we plan to integrate more data[60,61] to investigate the associations.

The significance of our research lies in two aspects. First, the existing guidelines for managing neonatal hyperbilirubinemia do not account for the link between air pollution exposure and the risk of neonatal jaundice. In 2004, the American Academy of

Pediatrics (AAP) made a time-bilirubin curve and a follow-up jaundice scheme after discharge. They examined the possible risk factors that caused neonatal jaundice. Yet, they failed to assess high risks of neonatal jaundice attributable to newborn exposure to air pollution. In China, the Society of Pediatrics of Chinese Medical Association (SPCMA) did not associate air pollution exposure with neonatal jaundice either. Through evaluating the impacts of air pollution exposure on neonatal jaundice incidence, our study provides evidence that air pollution exposure is a high risk factor for neonatal jaundice incidence. We suggest the AAP and SPCMA to improve the guidelines through taking air pollution exposure as a risk factor of newborn jaundice.

A second importance lies in air pollution exposure monitoring and pollution mitigation for newborns. Millions of infants are born in China each year. In 2016, China fully liberalized the second child policy, which led to a significant increase in the number of newborns over the last 2 years. Despite recent stringent control measures, air pollution in China remains severe and in 2014 only 7% of its population lived at places with the $PM_{2.5}$ concentration below the current annual China National Ambient Air Quality Standard[5]. At the very least, it is very important to strengthen the follow-up of neonatal jaundice in low air quality environments and closely observe newborns' bilirubin levels to ensure timely treatment against the harm of high bilirubin to them. To avoid air pollution exposure, measures to reduce air pollution in obstetric wards are necessary.

## Methods

**Neonatal and maternal clinical data.** We collected the dataset of 44,029 newborns born in Beijing from June 2014 through May 2017. Among these newborns, 2349 were premature, and 14,926 were discharged in 1–2 days after birth so that their bilirubin levels were not regularly monitored. These newborns were excluded from the dataset. We also removed 972 newborns who were admitted to the neonatal ward due to neonatal asphyxia, neonatal hemolytic disease, or neonatal aspiration pneumonia. We finally selected 25,782 term singleton newborns without hemolytic disease in this study. These newborns stayed in the wards all the time before they were discharged from the hospital.

For each newborn, we collected his/her mother's information such as age, occupation, educational level, gravidity, gestational age, pregnancy complications, delivery data (delivery time, delivery mode, postpartum hemorrhage, intrapartum hemorrhage, bleeding reason, and emergency rescue), labor time (I, II, III, assembly, complications), as well as other clinical monitoring items such as number of white blood cells, neutrophil percentage (GR), umbilical blood flow (S per D) and blood pressure. Neonatal characteristics included gender, height, weight, Apgar Score, infant special cases, and the delivery process like fetal distress, umbilical cord, and amniotic fluid.

**Air pollution and meteorological data.** We obtained the hourly air pollution data over June 2014–May 2017 from 34 air pollution monitoring stations of the MEE. Pollutant species included $PM_{10}$, $PM_{2.5}$, $SO_2$, CO, $NO_2$ and $O_3$. The $NO_2$ measurements were contaminated by other nitrogen species due to limitations of the measurement method[62]; the data were thus excluded here. We also excluded the $PM_{10}$ data due to large amount of missing values.

We obtained 3-hourly meteorological data over June 2014–May 2017 from one station near the southwest fourth ring road of Beijing. Data at this station are reported to the World Meteorological Organization and maintained at the United States National Oceanic and Atmospheric Administration National Centers for Environment Information (NOAA NCEI). Three-hourly data of atmospheric visibility, air temperature (2m) and relative humidity were collected. The cloud data contained too many missing values and were thus excluded.

We filled the missing meteorological or air pollution data through three linear interpolation methods in the preference order. The primary interpolation method (Eq. 1) took into account both diurnal and day-to-day variations, and it was applied when the following conditions were satisfied: for the hour $t$ on day $d$ with missing value, there are valid data at hour $t$, $t - \Delta t$ and $t + \Delta t$ (where $\Delta t \leq 6$) on day $d + \Delta d$ and $d - \Delta d$ (where $\Delta d \leq 3$). Valid data on paired days ($d + \Delta d$ and $d - \Delta d$) closest to $d$ were chosen.

$$\begin{aligned} \Delta data1 &= data(i, t, d - \Delta d) - (data(i, t - \Delta t, d - \Delta d) + data(i, t + \Delta t, d - \Delta d))/2 \\ \Delta data2 &= data(i, t, d + \Delta d) - (data(i, t - \Delta t, d + \Delta d) + data(i, t + \Delta t, d + \Delta d))/2 \\ \Delta data &= (\Delta data1 + \Delta data1)/2 \\ data(i, t, d) &= (data(i, t - \Delta t, d) + data(i, t + \Delta t, d))/2 + \Delta data \end{aligned} \quad (1)$$

where $i$ represents a parameter (air pollutant or meteorological variable).

The second interpolation method was a less stringent version of the first method. It was applied when there were no adequate pairs of days ($d + \Delta d$ and $d - \Delta d$) with valid data. Here, the closest day within $\Delta d$ of $d$ with valid data at hour $t$, $t - \Delta t$ and $t + \Delta t$ was used to do the interpolation.

The third method used the average of all valid data at hour $t$ in the closest 6 days of $d$ to fill the missing data at hour $t$ on day $d$.

For each site of $PM_{2.5}$, $SO_2$ and CO measurements in Beijing, 94–96% of hourly data were available, i.e., without the need of interpolation. The amount of air pollution data interpolated by the first and second methods together did not exceed 3%, and those interpolated by the third method did not exceed 5%. For air temperature and relative humidity, 97% of three-hourly data were available, i.e., without the need of interpolation. The amount of air temperature and relative humidity data interpolated by the first and second methods together did not exceed 2%, and those interpolated by the third method did not exceed 4%.

**Statistical analysis.** The logistic regression model was employed to assess the association between the risk of neonatal jaundice and air pollution, mother's age, sex, birth weight, gestational age, hypertension in pregnancy, gestational diabetes, fetal distress in uterus, cord around neck, premature rupture of membranes, infection, hypothyroidism, and anemia during pregnancy. Under the assumption of a two-sided alternative hypothesis, the P value <0.05 was considered to indicate statistical significance.

**Associating neonatal exposure and the bilirubin levels.** We used data at the Nongzhanguan and Dongsihuan air quality monitoring stations that are nearest to the hospital for neonatal exposure. Air pollutant data at these two stations were averaged, using weights inverse to their distances to the hospital. Using the inverse distance weighted average of all 34 stations led to a similar result in terms of the jaundice−pollution relationship.

Through the Kolmogorov−Smirnov test, we found that the peak bilirubin levels of newborns approximately exhibited a normal distribution. We thus utilized the GAM to explore the quantitative impacts of individual non-linearity related with air pollution issues[63,64]. It is a regression model in which smoothing splines are utilized for covariates[65,66].

We constructed the GAM in Eq. 2 to evaluate the relationship between air pollutants ($PM_{2.5}$, $SO_2$ and CO) and the peak bilirubin levels. The detailed health information of newborns, air temperature (degree Celsius) and relative humidity (%) obtained from the meteorological station were used as confounding factors.

In Eq. 2, the additive items for the three pollutants were constructed and the spline smoothing items were used to control the impacts of air pollutants on the bilirubin level.

$$\begin{aligned} g(u) = \ & \beta_0 + s(PM_{2.5}, df_1) + s(SO_2, df_2) + s(CO, df_3) + \lambda_1(rh) + \lambda_2(tem) \\ & + \lambda_3(ph) + \lambda_4(gd) + \lambda_5(fd) + \lambda_6(pr) + \lambda_7(ms) + \lambda_8(uc) + \lambda_9(ip) \\ & + \lambda_{10}(hy) + \lambda_{11}(an) + \varepsilon_i, \end{aligned} \quad (2)$$

where $u = E(Y|x_1, x_2, …, x_p)$, is the mathematical expectation of the independent variable $x$ ($PM_{2.5}$, $SO_2$, or CO). rh is the daily relative humidity. tem is the daily mean temperature. $s$ is a nonparametric smoothing function. $df_i$ represents the degree of freedom and is used to control the impact of the pollutants on bilirubin levels. The final degree of freedom of each variable is evaluated according to Akaike's Information Criteria (AIC)[67]. The degree of freedom of each variable is adjusted to minimize the AIC. $\lambda_1, \lambda_2, …, \lambda_{11}$ are parameters. ph, gd fd, pr, ms, uc, ip, hy and an are binary classification variables representing the influences of maternal hypertension in pregnancy, gestational diabetes, fetal distress in uterus, premature rupture of membranes, meconium-stained amniotic fluid, umbilical cord around neck, infection during pregnancy, hypothyroidism and anemia on neonatal jaundice risks, respectively.

The fitted GAM is as follows:

$$\begin{aligned} g(u) = \ & 11.39 + 0.04(PM_{2.5}) + s(PM_{2.5}) + 0.43(SO_2) + s(SO_2) \\ & + 0.90(CO) + 0.98s(CO) - 0.003(rh) + 0.001(tem) + 0.25(fd) \\ & + 0.18(ip) + 0.12(uc) + 0.13(ph) \end{aligned} \quad (3)$$

**Associating maternal exposure and the bilirubin level.** We estimated the maternal exposure to air pollutants using the air pollution monitoring station located nearest to each mother's residence address. The GAM (Eq. 4) was used to further assess the relationships between maternal exposure to pollutants in the third trimester of pregnancy and the peak bilirubin level of her newborn.

$$\begin{aligned} g(u) = \ & \beta_0 + s(PM_{2.5}, df_1) + s(SO_2, df_2) + s(CO, df_3) + \lambda_1(rh) \\ & + \lambda_2(tem) + \lambda_3(ph) + \lambda_4(gd) + \lambda_5(fd) + \lambda_6(pr) + \lambda_7(ms) \\ & + \lambda_8(uc) + \lambda_9(ip) + \lambda_{10}(hy) + \lambda_{11}(an) + \varepsilon_i \end{aligned} \quad (4)$$

The fitted result of Eq. 4 is as follows:

$$g(u) = 10.35 + 0.12(PM_{2.5}) + s(PM_{2.5}) + 0.22(SO_2) + s(SO_2)$$
$$+ 1.11CO + 0.97s(CO) - 0.003(rh) + 0.006(tem) + 0.26(fd) \qquad (5)$$
$$+ 0.17(ip) + 0.14(uc) + 0.15(ph)$$

**Linking neonatal exposure time to the bilirubin level**. We used the geographical detector[68] to quantify the individual influences of average pollutant concentration (average of daily air pollution from the day of birth to the day before the peak bilirubin level was measured) and exposure time (in hours) on the peak bilirubin level of a newborn. The geographical detector has the ability to detect the extent to which the determinant $x$ can explain the variability of the attribute $y$. Here, let $q$ be a measure of the calculation results, and its value range is [0, 1]. The larger $q$ is, the stronger the explanatory power of $x$ to $y$ is.

The exposure time was divided into six strata, 0–24, 24–48, 48–72, 72–96, 96–120 and >120 h. The effect of $PM_{2.5}$, $SO_2$ and CO exposure time was examined separately. The $PM_{2.5}$ average concentration was divided into five strata, 0–60, 60–96, 96–153 and >153 μg per m$^3$. The resulting exposure time's $q$ value was 0.223, and the average $PM_{2.5}$ concentration's $q$ value was 0.00902. Their $P$ values were both less than 0.05; thus they were explanatory variables of the peak bilirubin level. When the exposure time and average $PM_{2.5}$ concentration were combined, $q$ reaches a value of 0.243, only marginally larger than the sum of above two $q$ values. Thus the positive interaction between exposure time and average $PM_{2.5}$ concentration was weak.

The effects of $SO_2$ and CO exposure time were analyzed in a similar way. The exposure time stratification was the same as above. The $SO_2$ average concentration was divided into four strata, 0–3.54, 3.54–5.96, 5.96–9.42 and >9.42 μg per m$^3$. The $q$ value of $SO_2$ concentration was 0.00154 and the corresponding exposure time was 0.223. The interaction detection showed that they had a weak positive interaction, and the combined $q$ value was 0.241. The CO average concentration was divided into four strata according to the quartiles: 0–0.68, 0.68–1.08, 1.08–1.55 and >1.55 mg per m$^3$. The q value of CO concentration was 0.0134 and the corresponding exposure time was 0.223. The interaction detection found that they had a weak positive interaction, and the combined $q$ value was 0.248.

We further constructed a GAM for the exposure time, average pollutant concentration and their interaction as follows:

$$g(u) = s(x_i) + s(t) + s(x_i, t), \qquad (6)$$

where $s(x_i, t)$ denotes how the interaction between average pollutant concentration and exposure time affects the peak bilirubin level. $x_i$ denotes a certain air pollutant $i$, i.e., $x_1$ for $PM_{2.5}$, $x_2$ for $SO_2$, and $x_3$ for CO.

**Sensitivity analysis**. In this study population, the newborns stayed in the wards all the time before they were discharged from the hospital, and thus they were not affected by outdoor sunlight. Here we evaluated whether TOA solar irradiance affected the relationship between air pollution exposure and neonatal jaundice by stratified analysis of daily TOA solar irradiance (controlling for visibility and other confounding factors). The daily TOA solar irradiance was calculated based on the solar constant, latitude, date, and solar hour angle. It was categorized into four levels separated by the 25%, 50% and 75% percentiles of the daily TOA solar irradiance intensities: below 252.1, 252.1–283.8, 283.8–313.2, and above 313.2 w per m$^2$, respectively. Results are shown in Supplementary Table 8.

We evaluated whether visibility affected the relationship between air pollution exposure and neonatal jaundice by stratified analysis of daily visibility (controlling for TOA irradiance and other confounding factors). Atmospheric visibility was categorized into four levels separated by the 25%, 50% and 75% percentiles of the atmospheric visibility range: below 4.7, 4.7–8.2, 8.2–15.9, and above 15.9 km, respectively. Results are shown in Supplementary Table 9.

To test whether the influences of the three factors, i.e. air pollution exposure time of newborns, air pollutant concentration newborns were exposed to, and maternal exposure to air pollution during the third trimester, on the risk of neonatal jaundice were independent of each other, we constructed the GAM model (Eq. 7) combining the three factors for sensitivity analysis.

$$g(u) = \beta_0 + s(mX_i, df_1) + s(bX_i, df_2) + s(Time_i) + \varepsilon_j, \qquad (7)$$

where $mX_i$ denotes the average concentration of maternal exposure to the $i$th air pollutant in the third trimester of pregnancy. $bX_i$ denotes the average concentration of the $i$th air pollutant a newborn was exposed to. $Time_i$ denotes the time of the neonatal exposure to the $i$th air pollutant. $I = 1$ denotes $PM_{2.5}$, $i = 2$ denotes $SO_2$, and $i = 3$ denotes CO.

In the GAM model, we assessed the association of each factor with the risk of neonatal jaundice through controlling the other two factors. The estimated results were illustrated in Supplementary Figs. 3–5.

**Human subjects research statement**. This study was approved by the Institutional Review Boards/Ethics Committees of Beijing Obstetrics and Gynecology Hospital. It complied with the Declaration of Helsinki. Verbal assent and written consent were obtained from all study subjects and their parents in accordance with HIPAA regulations prior to partaking in the study.

## Data availability

The collected neonatal sternal skin images and any remaining data can be available from the corresponding authors on reasonable request.

## Code availability

The source code can be available from the corresponding authors upon reasonable request. It is copyrighted by Beijing Normal University, and Beijing Obstetrics and Gynecology Hospital and is to be used only for educational and research purposes. Any commercial use is prohibited.

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

## Acknowledgements

This work was carried out with the support of the National Key Research and Development Program of China under Grant 2018YFC0213600, the National Natural Science Foundation of China under Grant 41775115 and Grant 41371324, and the Beijing Natural Science Foundation under Grant 7173258.

## Author contributions

Liqiang Z. developed the research framework, analyzed the results, and wrote/edited the paper. W.L. contributed to the goal of the research as well as collection and analysis of neonatal and maternal data. K.H. contributed to the model of the paper. J.L. developed the research framework, provided air pollution and meteorological data, contributed to results analysis, and edited the paper. C.S., C.Z. and B.H. improved the research framework and edited the paper. X.T., J.W. and W.R. gave important suggestions for improving the paper. Y.J. organized the neonatal and maternal datasets. Ziwei W., N.R., M.L., Liang Z. and Ziye W. generated the air pollution and meteorological datasets. Yuebin W., X.L., S.L. and Yanhong W. developed the maps and edited figures.

## Additional information

**Competing interests:** The authors declare no competing interests.

