## [Peer Review File · Nature Communications]

Reviewers' comments:

Reviewer #1 - expert in neonatal jaundice (Remarks to the Author):

General

The authors of this manuscript set out to evaluate the potential impacts of air pollution exposure on the risk of neonatal jaundice as well as the magnitude and mechanisms of these impacts among term singleton newborns in one maternity hospital in China.

Two basic strands of association were explored:

1. The relationship between maternal exposure to air pollution during pregnancy and the incidence of jaundice in the newborns
2. The risk of jaundice in newborns exposed to air pollution.

The following findings were reported:

1. A significant association between overall air quality (AQI) and the incidence of neonatal jaundice (defined as peak bilirubin level of 5 mg/dL or more).
2. Significant association between different air pollutants with the risk of clinically significant jaundice (defined as peak bilirubin level of 13mg/dL or more).
3. The risk of neonatal jaundice increased linearly with duration of exposure to specific air pollutants from birth to 48 hours postnatal age but slowed down between 48 to 120 hours of exposure.
4. Maternal exposure to specific air pollutants during the third trimester of pregnancy was significantly associated with the risk of clinically significant jaundice.
5. Two possible mechanisms were proposed for the reported impact of neonatal exposure to air pollution and the risk of neonatal jaundice, namely:
 - a. The effect of exposure to direct sunlight on bilirubin levels
 - b. The effect of breathed air pollutants on liver function.

Overall, the study is potentially an interesting complement to the vast literature on the adverse perinatal outcomes associated with maternal and neonatal exposure to air pollutants. However, I have several serious methodological concerns in relation to the determination of jaundice in the

newborns that will limit the validity of this study. Specifically, the following observations need to be addressed.

Introduction

1. Page 2, line 51: the inclusion of references 17 and 18 would suggest that prior studies have identified neonatal jaundice as an adverse perinatal outcome from air pollution which contradicts the authors' claim of novelty in lines 61 and 62.
2. Page 2, lines 57 and 58: The statement that neonatal jaundice has been a priority for WHO based on a 1985 report is most inaccurate. This reviewer is not aware of any WHO document that considers neonatal jaundice as a global health priority.
3. The sentence listing risk factors for neonatal jaundice should be supported with appropriate references.
4. Although studies specifically exploring the association between air pollutants and neonatal jaundice are rare, there are studies on the perinatal outcomes of exposure to tobacco smoke where neonatal jaundice is mentioned (e.g. Crane et al. Effects of environmental tobacco smoke on perinatal outcomes: a retrospective cohort study. *BJOG* 2011;118:865–871).

Methodology

5. The rationale for exploring the impact of air pollution on the incidence of benign physiological jaundice which affects the vast majority of newborns worldwide is difficult to appreciate. A more desirable goal should have been to determine the contribution of exposure to air pollutants to the risk of clinically significant jaundice requiring treatment. It is noteworthy that none of the infants included in this study received phototherapy which is an indication of the severity of jaundice in the study population.
6. It is unclear how many bilirubin measurements were obtained for each infant and the interval between measurements in relation to the time of discharge to determine the peak bilirubin level used in this study.
7. The entire proposition on the role of sunlight exposure to the incidence of jaundice in this study population in a hospital located outside the tropics is counter-intuitive. How was the irradiance level in the ward determined to account for the impact of sunlight?
8. It is unclear how and why babies in the same ward had different exposure to air pollution.
9. Several references have inaccurate citation details or are not appropriate for related statements.

Overall, the evidence presented in this study suggesting that air pollution should be considered as a risk factor in the management guidelines for neonatal jaundice is extremely weak and fundamentally unreliable without comparing babies exposed and not exposed to air pollution in properly randomized control study.

Reviewer #2 - expert in epidemiology and health risks associated with air pollution (Remarks to the Author):

This is a well conducted study examining the impact of air pollution on neonatal jaundice in Beijing, China. The main contribution of this paper is elucidating the impact of breathed air pollution on Jaundice risk (not examining air pollution as a surrogate for lower sunlight exposure) and while the authors attempt to control for sunlight exposure this should be improved. Below are specific comments.

- Figure 1 illustrates the spatial patterns of PM but the article focusing on the temporal component in analyses. A graph of the temporal air pollution exposure levels, together with estimates of TOA and atmospheric visibility should be added here. This will provide further information on whether air pollution is representing lower sunlight exposure or if there could be a separate contribution from breathed air pollution.
- It is unclear how the spatial concentrations of air pollutants were assigned to mother's residential address (for the 3rd trimester exposure analyses). Is this based on the hospital monitors?
- The difference in pollution exposure for Jaundice and non-jaundice babies is extremely large, and likely suggests a seasonal component (i.e. higher air pollution concentrations in the winter). The analysis attempts to control for solar irradiance using a TOA covariate, but further adjusting for week should be added to the main model or in sensitivity analysis.
- The fact that Ozone was not associated with Jaundice further supports that these results may be driven by season, as Ozone is typically higher in summer months. These results, should still be presented in the manuscript even if not statistically significant.
- The correlation between different air pollution exposures should be presented. Do these represent unique exposures or are they highly correlated and reflect the air pollution mix on bad air quality days? Are all pollutants equally correlated with atmospheric visibility? If not this would provide another option for pulling apart the influence of breathed air pollution and sunlight exposures.
- A major contribution of this study is elucidating the impact of breathed air pollution on Jaundice risk and the analyses should therefore carefully control for sunlight exposure (impacted by season and atmospheric visibility) in the analysis of breathed air pollution risk. Stratified analysis by TOA and atmospheric visibility would help here.

- Table 1 does not include any socio-demographic characteristics. Are these available in the health records? The fact that there was such a strong difference in hypertension in pregnancy is surprising.

Response to Reviewers' Comments for Manuscript

NCOMMS-18-30209

Air pollution exposure and neonatal jaundice

Liqiang Zhang, Weiwei Liu, Jintai Lin, Kun Hou, Chenghu Zhou, Bo Huang, Xiaohua Tong, William Rhine, Jinfeng Wang, Ying Jiao, Ziwei Wang, Ruijing Ni, Mengyao Liu, Liang Zhang, Ziyue Wang, Yuebin Wang, Yanhong Wang

Dear Reviewers,

We would like to thank you for your valuable comments that helped us revise and improve the presentation and the technical context of our manuscript. We are also encouraged by the positive comments made by you regarding the usefulness and novelty of our research. In our individual replies, we have addressed all of the comments and suggestions. The revised main manuscript and Supplementary Material now includes appropriate modifications that have been made possible from your very thoughtful comments. Extensive revisions have been made throughout the main manuscript and Supplementary Material as follows.

- 1) Presentations of our manuscript have been modified and improved carefully.
- 2) Some related references have been added, and the inaccurate citation details and improper related statements have been corrected in the main manuscript.
- 3) We compared babies exposed and not exposed to air pollution in properly randomized control study.
- 4) The graphs of the temporal air pollution exposure levels, together with estimates of TOA and atmospheric visibility were provided.
- 5) The correlations between different air pollution exposures as well as between single air pollutant and atmospheric visibility were analysed.
- 6) In this study population, the newborns stayed in the wards all the time before they were discharged from the hospital, and thus they were not affected by outdoor sunlight. We found that the associations between TOA solar irradiance/atmospheric visibility and neonatal jaundice were not statistically significant.
- 7) We accounted the week effect in the GAM model. Tables R5-R7 estimate the associations of neonatal exposure to air pollutants (PM_{2.5}, SO₂ and CO) with the

peak bilirubin levels after/before controlling for week. From the tables, we note that week effect has little influences on the results regarding the pollution-jaundice relationship.

- 8) Table R1 includes more socio-demographic characteristics obtained from the healthrecords.

The modifications made in the revised main manuscript and Supplementary Material are highlighted using a yellow background.

In the following, we have addressed all of your comments and suggestions. Unless otherwise stated, the references given in our replies can be found in the revised main manuscript.

Response to Reviewer 1

Question: *Overall, the study is potentially an interesting complement to the vast literature on the adverse perinatal outcomes associated with maternal and neonatal exposure to air pollutants. However, I have several serious methodological concerns in relation to the determination of jaundice in the newborns that will limit the validity of this study. Specifically, the following observations need to be addressed.*

Our reply: Thanks very much for the positive comments. We have addressed all of your comments, and correspondingly revised the main manuscript and Supplementary Material.

Question: *Page 2, line 51: the inclusion of references 17 and 18 would suggest that prior studies have identified neonatal jaundice as an adverse perinatal outcome from air pollution which contradicts the authors' claim of novelty in lines 61 and 62.*

Our reply: In our original manuscript, the reference 17 “Ronald, S. *et al.* Understanding Neonatal Jaundice: A Perspective on Causation. *Pediatr Neonatol*, 51(3), 143–148 (2010)” discussed the role of Carbon Monoxide (CO) in Neonatal Jaundice, yet it failed to estimate the quantitative correlation between maternal or neonatal exposure to CO as well as between other air pollutants (PM_{2.5} and SO₂) and the incidence of neonatal jaundice. The reference 18 “Akinpelu, O. V. *et al.* Auditory risk of hyperbilirubinemia in term newborns: A systematic review. *International Journal of Pediatric Otorhinolaryngology*, 77(6), 898-905 (2013)” mainly gave a systematic review of clinical studies to evaluate the effect of hyperbilirubinemia on hearing in term newborns. The aim of the reference was to find the relationship between hearing function and bilirubin levels as well as the effect of treatment, but

did not link air pollution exposure to the neonatal jaundice incidence. Both of the references did not quantify the associations between the incidence of neonatal jaundice and air pollution.

The novelty of our study mainly lies that we quantified the relationship between air pollutants (PM_{2.5}, SO₂ and CO) and the neonatal jaundice risk using data of 25,782 term singleton newborns without hemolytic disease, their mothers, air pollution and meteorological conditions across four years.

Question: *Page 2, lines 57 and 58: The statement that neonatal jaundice has been a priority for WHO based on a 1985 report is most inaccurate. This reviewer is not aware of any WHO document that considers neonatal jaundice as a global health priority.*

Our reply: In the main manuscript, we have improved the presentation, i.e. “Olusanya et al.²⁴ reported that the leading health-care policy research groups like the Child Health Epidemiology Reference Group of the World Health Organization (WHO) and the Global Burden of Disease Collaborators increasingly recognize the clinical and public health significance of neonatal jaundice or hyperbilirubinemia as an important neonatal condition that deserves global health attention in the post-2015 millennium development goals era^{25,26}”.

Please see the last paragraph on page 2 in the revised main manuscript.

Question: *The sentence listing risk factors for neonatal jaundice should be supported with appropriate references.*

Our reply: The related references have been given in the listing risk factors for neonatal jaundice. Please see the last paragraph on page 2 in the revised main manuscript.

Question: *Although studies specifically exploring the association between air pollutants and neonatal jaundice are rare, there are studies on the perinatal outcomes of exposure to tobacco smoke where neonatal jaundice is mentioned (e.g. Crane et al. Effects of environmental tobacco smoke on perinatal outcomes: a retrospective cohort study. BJOG 2011;118:865–871).*

Our reply: We have carefully studied the article of Crane *et al.*^[31] and other related articles^[32]. Indeed, exposure of pregnant women to environmental tobacco smoke

was associated with the risk of neonatal jaundice; the relationship between air quality and the neonatal jaundice risk remains poorly quantified. The toxicity of smoking may differ greatly from that of ambient air pollution. In this study, we used more data (including air pollution, neonatal data and pregnant women data) to quantify the dose–response relationships.

Question: *The rationale for exploring the impact of air pollution on the incidence of benign physiological jaundice which affects the vast majority of newborns worldwide is difficult to appreciate. A more desirable goal should have been to determine the contribution of exposure to air pollutants to the risk of clinically significant jaundice requiring treatment. It is noteworthy that none of the infants included in this study received phototherapy which is an indication of the severity of jaundice in the study population.*

Our reply: We are sorry that we did not describe the presentation clearly. In lines 69 and 70 of the original manuscript, the presentation “They all stayed in the hospital from the birth to the data collection, and did not receive phototherapy for the treatment of hyperbilirubinemia.” means that for the jaundiced newborns who required treatment, we measured their bilirubin levels before they received phototherapy. The reason is that their bilirubin levels would decrease during the phototherapy.

The study population contains newborns without jaundice, physiological jaundice and serious jaundice (and thus need to receive phototherapy). Among the 14,058 jaundiced newborns, 7,722 experienced physiological neonatal jaundice, which were not intervened; whereas 6,336 had the risk of clinically significant jaundice and required close monitoring and prompt treatments according to the Chinese clinical guideline of neonatal jaundice (please refer to the references [1] and [2] at the end of this document) as shown in Table R1 (Supplementary Appendix Table S3). For the 6,336 newborns, we listed the number of jaundiced newborns who were considered phototherapy (Those who were considered to receive phototherapy usually received phototherapy. If they did not receive phototherapy, their bilirubin levels should be closely monitored. Once their bilirubin levels reached the degree of the required phototherapy, they must receive phototherapy) and required phototherapy (Those who were required to receive phototherapy actually received phototherapy) as shown in Table R2 (Supplementary Appendix Table S4). Therefore, there were lots of infants included in this study who received phototherapy in the study population.

Table R1. Recommended standards of neonatal jaundice intervention for full-term newborns of different birth days (please refer to the references [1] and [2] at the end of this document)

Days after birth	TSB levels (mg/dL)	
	Jaundice considering phototherapy	Jaundice requiring phototherapy
≤ 1	≥ 6	≥ 9
2	≥ 9	≥ 12
3	≥ 12	≥ 15
>3	≥ 15	≥ 17

Table R2. Jaundiced newborns who were considered phototherapy or required phototherapy

Days after birth	no. of newborns (TSB levels, mg/dL)	
	Jaundice considering phototherapy	Jaundice requiring phototherapy
≤ 1	426 (≥ 6)	736 (≥ 9)
2	573 (≥ 9)	934 (≥ 12)
3	891 (≥ 12)	1,152 (≥ 15)
>3	667 (≥ 15)	957 (≥ 17)
Sum of newborns	2,557	3,779

Question: *It is unclear how many bilirubin measurements were obtained for each infant and the interval between measurements in relation to the time of discharge to determine the peak bilirubin level used in this study.*

Our reply: For the newborns whose bilirubin levels were far less than the level needing treatment, we usually measured their bilirubin levels every six hours. For those who would approach the levels of the considering phototherapy, we measured their bilirubin levels every one hour before they received phototherapy.

Question: *The entire proposition on the role of sunlight exposure to the incidence of jaundice in this study population in a hospital located outside the tropics is counter-intuitive. How was the irradiance level in the ward determined to account for the impact of sunlight?*

Our reply: As clarified in the revised manuscript, in this study population, the newborns stayed in the wards all the time before they were discharged from the hospital, and thus they were not affected by outdoor sunlight.

As a support to this, below shows that TOA irradiance had no influences on the relationship between air pollution exposure and neonatal jaundice. We estimated the association between air pollution exposure and neonatal jaundice by stratified analysis of daily TOA solar irradiance (controlling for other confounding factors, Supplementary Appendix Table S10). Daily TOA solar irradiance was categorized into four levels separated by the 25%, 50% and 75% percentiles of the daily TOA solar irradiance intensities: below 252.1 w/m², 252.1-283.8 w/m², 283.8-313.2 w/m², and above 313.2 w/m². We found the association between air pollution exposure and neonatal jaundice to be quantitatively similar at different TOA percentiles. Thus, the air pollution-jaundice relationship does not depend on the level of TOA irradiance.

Question: *It is unclear how and why babies in the same ward had different exposure to air pollution.*

Our reply: There were no air cleaning facilities in the wards, thus babies in the wards were exposed to ambient air pollution through indoor-outdoor air exchange. Different babies stayed in the wards at different times and thus were exposed to different levels of air pollution.

Question: *Several references have inaccurate citation details or are not appropriate for related statements.*

Our reply: We have proofread the whole manuscript and modified the inaccurate citation details and improper related statements accordingly.

Question: *Overall, the evidence presented in this study suggesting that air pollution should be considered as a risk factor in the management guidelines for neonatal jaundice is extremely weak and fundamentally unreliable without comparing babies exposed and not exposed to air pollution in properly randomized control study.*

Our reply: On page 2, lines 115-122 of the original manuscript, we had compared the newborns exposed and not exposed to air pollution, i.e. “We used the logistic regression to examine whether air pollution exposure was associated with the risk neonatal jaundice. Among the 25,782 newborns, 15,228 were in the good air environment (AQI < 100, average of daily pollution from the day of birth to the day before the peak bilirubin level was measured), of which 6,200 (40.7%) had jaundice; and 10,554 were in poor air environment (AQI > 100), of which 7,858 (74.5%) had jaundice. The much higher percentage of neonatal jaundice (74.5% versus 40.7%; $\chi^2 =$

2862.105, P = 0.000) further supported the linkage between air pollution exposure and incidence of jaundice.”

We have further compared babies exposed and not exposed to air pollution. According to the ambient air quality criteria - global update 2005 proposed by World Health Organization (WHO), a 24-hour average concentration of 25 $\mu\text{g}/\text{m}^3$ was chosen as the short-term guideline value for $\text{PM}_{2.5}$, so we regard the newborns that were not exposed to air pollution as the 24-hour average $\text{PM}_{2.5}$ concentration was not larger than 25 $\mu\text{g}/\text{m}^3$. We estimated the associations of $\text{PM}_{2.5}$ with the peak bilirubin levels for $\text{PM}_{2.5}$ concentrations $\in (0, 25] \mu\text{g}/\text{m}^3$ and larger than 25 $\mu\text{g}/\text{m}^3$, respectively. From Table R3 (Supplementary Appendix Table S7), we note that the association between $\text{PM}_{2.5}$ and neonatal jaundice was not statistically significant for $\text{PM}_{2.5}$ concentrations $\in (0, 25] \mu\text{g}/\text{m}^3$ whereas the association was statistically significant for $\text{PM}_{2.5}$ concentrations above 25 $\mu\text{g}/\text{m}^3$. Some countries might decide to adopt lower concentrations than the WHO guideline values as their national air quality standards. In fact, Table 1 in the main manuscript also listed the association for $\text{PM}_{2.5}$ concentrations $\in (0, 10] \mu\text{g}/\text{m}^3$. Obviously, the association was not statistically significant.

Table R3. Association of $\text{PM}_{2.5}$ exposure with the peak bilirubin levels on the basis of an increase of 1.0 $\mu\text{g}/\text{m}^3$ in exposure to $\text{PM}_{2.5}$

$\text{PM}_{2.5}$ exposure intervals ($\mu\text{g}/\text{m}^3$)	Estimated risk in peak bilirubin levels (mg/dL)	Confidence lower limit (mg/dL)	Confidence upper limit (mg/dL)	P value
(0, 25]	0.005	-0.108	0.118	0.932
> 25	0.034	-0.005	0.073	0.008

According to the ambient air quality criteria, a 24-hour average concentration of 20 $\mu\text{g}/\text{m}^3$ was chosen as the short-term guideline value for SO_2 . Some countries might decide to adopt lower concentrations than the WHO guideline values as their national air quality standards. We estimated the associations of SO_2 with the peak bilirubin levels. From Table R4 (Supplementary Appendix Table S8), we note that the association between SO_2 and neonatal jaundice was not statistically significant for SO_2 concentrations $\in (0, 10] \mu\text{g}/\text{m}^3$.

Table R4. Association of SO₂ exposure with the peak bilirubin levels on the basis of an increase of 1.0 µg/m³ in exposure to SO₂

SO ₂ exposure intervals (µg/m ³)	Estimated risk in peak bilirubin levels (mg/dL)	Confidence lower limit (mg/dL)	Confidence upper limit (mg/dL)	P value
(0, 5]	0.082	-0.157	0.321	0.327
(5, 10]	0.028	-0.113	0.17	0.776
(10, 15]	0.094	0.077	0.111	< 0.001
>15	0.161	0.07	0.252	< 0.001

Please see the revisions in Supplementary Results in Supplementary Material.

Response to Reviewer 2

Question: *This is a well conducted study examining the impact of air pollution on neonatal jaundice in Beijing, China. The main contribution of this manuscript is elucidating the impact of breathed air pollution on Jaundice risk (not examining air pollution as a surrogate for lower sunlight exposure) and while the authors attempt to control for sunlight exposure this should be improved.*

Our reply: Thanks very much for the positive comments. As clarified in the revised manuscript, in this study population, the newborns stayed in the wards all the time before they were discharged from the hospital, and thus they were not affected by sunlight.

As a support to this, below shows that TOA irradiance had no influences on the relationship between air pollution exposure and neonatal jaundice. We estimated the association between air pollution exposure and neonatal jaundice by stratified analysis of daily TOA solar irradiance (controlling for other confounding factors, Supplementary Appendix Table S10). Daily TOA solar irradiance was categorized into four levels separated by the 25%, 50% and 75% percentiles of the daily TOA solar irradiance intensities: below 252.1 w/m², 252.1-283.8 w/m², 283.8-313.2 w/m², and above 313.2 w/m². We found the association between air pollution exposure and neonatal jaundice to be quantitatively similar at different TOA percentiles. Thus, the air pollution-jaundice relationship does not depend on the level of TOA irradiance.

Question: *Figure 1 illustrates the spatial patterns of PM but the article focusing on the temporal component in analyses. A graph of the temporal air pollution exposure levels, together with estimates of TOA and atmospheric visibility should be added*

here. This will provide further information on whether air pollution is representing lower sunlight exposure or if there could be a separate contribution from breathed air pollution.

Our reply: Supplementary Appendix Fig. S3 illustrates the monthly exposure levels of particulate matter with diameter below 2.5 micron (PM_{2.5}), sulfur dioxide (SO₂), ozone (O₃), and carbon monoxide (CO) of the 34 air quality monitoring sites in Beijing, together with estimates of TOA irradiance and atmospheric visibility. From this figure and Supplementary Appendix Fig. S7, we lower sunlight (winter) months also tend to have heavier pollution (except O₃). However, as clarified in the revised manuscript, in this study population, the newborns stayed in the wards all the time before they were discharged from the hospital, and thus they were not affected by outdoor sunlight. This point is already discussed in response to the last comment.

Question: *It is unclear how the spatial concentrations of air pollutants were assigned to mother's residential address (for the 3rd trimester exposure analyses). Is this based on the hospital monitors?*

Our reply: There are 34 stations in Beijing city operated by the Ministry of Ecology and Environment (formally the Ministry of Environmental Protection). We estimated the maternal exposure to air pollutants using the air pollution monitoring station located nearest to each mother's residence address. We have clarified this point in the revised Methods.

Question: *The difference in pollution exposure for Jaundice and non-jaundice babies is extremely large, and likely suggests a seasonal component (I.e. higher air pollution concentrations in the winter). The analysis attempts to control for solar irradiance using a TOA covariate, but further adjusting for week should be added to the main model or in sensitivity analysis.*

Our reply: Supplementary Appendix Fig. S1 illustrates average monthly air pollutant concentrations of the 34 air quality monitoring sites in Beijing. Opposed to other pollutants, ozone concentrations normally reach a minimum in winter. Also, the winter-heating in Beijing greatly enhanced emissions and ambient levels of non-ozone pollutants – in general, PM_{2.5} concentrations in the heating seasons were much higher than those in the non-heating seasons, which are consistent with the findings (please refer to the references [3] at the end of this document). As pointed out by the reviewer, the seasonal dependence was removed by controlling for TOA solar irradiance.

We further adjusted for week in the GAM model. The days from Monday to Saturday are set as the dummy variables, and Sunday as the reference category. They were added to the GAM model (Eq. 1). Tables R5-R7 estimate the associations of neonatal exposure to air pollutants (PM_{2.5}, SO₂ and CO) with the peak bilirubin levels after/before controlling for week. From the tables, we note that controlling for week has little influences on the results regarding the pollution-jaundice relationship.

$$g(u)=\beta_0+s(\text{PM}_{2.5},df_1)+s(\text{SO}_2,df_2)+s(\text{CO},df_3)+\lambda_1(rh)+\lambda_2(tem)+\lambda_3(ph)+\lambda_4(gd)+\lambda_5(fd)+\lambda_6(pr)+\lambda_7(ms)+\lambda_8(uc)+\lambda_9(ip)+\lambda_{10}(hy) +\lambda_{11}(an)+DOW+\varepsilon_i \quad (1)$$

where DOW=(Monday, ..., Saturday)

The fitted result of Eq. (1) is as follows:

$$g(u)=10.395+0.08(\text{PM}_{2.5})+s(\text{PM}_{2.5})+0.27(\text{SO}_2)+s(\text{SO}_2)+1.14\text{CO}+0.98s(\text{CO})-0.004(rh)+0.008(tem)+0.21(fd)+0.18(ip)+0.16(uc)+0.17(ph)+0.27(\text{Monday})+0.21(\text{Tuesday})+0.19(\text{Wednesday})+0.31(\text{Thursday})+0.29(\text{Friday})+0.02(\text{Saturday}) \quad (2)$$

Table R5. Associations of PM_{2.5} exposure with the peak bilirubin levels on the basis of an increase of 1.0 µg/m³ in exposure to PM_{2.5} after/before controlling for week

Exposure intervals (µg/m ³)	Estimated risk in peak bilirubin levels (95 % CI) after controlling for week (mg/dL)	Estimated risk in peak bilirubin levels (95 % CI) before controlling for week (mg/dL)
(0, 10]	0.826 (-0.596, 2.247)	0.848 (-0.574, 2.269)
(10, 35]	0.073 (0.024, 0.122)	0.076 (0.027, 0.125)
(35,75]	0.025 (0.010, 0.040)	0.029 ((0.014, 0.044)
(75, 200]	0.008 (0.001, 0.015)	0.009 (0.002, 0.016)
>200	0.010 (-0.008, 0.028)	0.010 (-0.008, 0.028)

Table R6. Associations of SO₂ exposure with the peak bilirubin levels on the basis of an increase of 1.0 µg/m³ in exposure to SO₂

Exposure intervals (µg/m ³)	Estimated risk in peak bilirubin levels (95 % CI) after controlling for week (mg/dL)	Estimated risk in peak bilirubin levels (95 % CI) before controlling for week (mg/dL)
(0, 5]	0.085 (-0.154, 0.324)	0.082 (-0.157, 0.321)
(5, 10]	0.032 (-0.109, 0.174)	0.028 (-0.113, 0.17)
(10, 15]	0.089 (0.072, 0.106)	0.094 (0.077, 0.111)
>15	0.163 (0.072, 0.254)	0.161 (0.07, 0.252)

Table R7. Associations of CO exposure with the peak bilirubin levels on the basis of an increase of 1.0 mg/m³ in exposure to CO

Exposure intervals (mg/m ³)	Estimated risk in peak bilirubin levels (95 % CI) after controlling for week (mg/dL)	Estimated risk in peak bilirubin levels (95 % CI) before controlling for week (mg/dL)
(0, 3.5]	0.348 (0.301, 0.375)	0.351 (0.314, 0.388)

Question: *The fact that Ozone was not associated with Jaundice further supports that these results may be driven by season, as Ozone is typically higher in summer months. These results, should still be presented in the manuscript even if not statistically significant.*

Our reply: The seasonal dependence was removed by controlling for TOA solar irradiance, which is greatly season-dependent (Supplementary Information Fig. S3). We analyzed the association between ozone and the risk of neonatal jaundice. The results are presented in Table R8 (Supplementary Appendix Table S5).

Table R8. Association between jaundice severity and concentrations of individual air pollutants

Concentrations of air pollutants	Physiological jaundice (n=7,722)	Clinically significant jaundice (n=6,336)	T value	P value
Mean PM _{2.5} concentration (µg/m ³)	105.68±94.44	123.30±96.00	-10.932	0.000
Mean SO ₂ concentration (µg/m ³)	9.56±7.43	11.92±9.21	-16.765	0.000
Mean O ₃ concentration (µg/m ³)	32.33±23.66	30.52±21.31	1.930	0.054
Mean CO concentration (mg/m ³)	1.55±1.40	1.64±1.18	-4.164	0.004
Max PM _{2.5} concentration (µg/m ³)	183.93±135.12	204.73±132.10	-9.178	0.000
Max SO ₂ concentration (µg/m ³)	19.88±15.04	23.90±16.10	-15.242	0.000
Max O ₃ concentration (µg/m ³)	82.66±57.40	79.91±56.17	1.162	0.246
Max CO concentration (mg/m ³)	2.68±2.01	2.90±1.83	-6.468	0.008

Question: *The correlation between different air pollution exposures should be presented. Do these represent unique exposures or are they highly correlated and reflect the air pollution mix on bad air quality days? Are all pollutants equally correlated with atmospheric visibility? If not this would provide another option for pulling apart the influence of breathed air pollution and sunlight exposures.*

Our reply: In the generalized additional model (GAM), we can separate the

influences of each air pollutant on another air pollutant or atmospheric visibility. We estimated the correlations between different air pollution exposures as listed in Table R9 (Supplementary Appendix Table S1) based on daily mean air pollution data over June 2014–May 2017 obtained from 34 air pollution monitoring stations. As listed in Table R9, PM_{2.5} was highly correlated with SO₂ and CO, whereas O₃ was negatively correlated with SO₂ and CO. They reflected the air pollution mix on bad air quality days. There was the greatest negative correlation between PM_{2.5} exposure and atmospheric visibility, mainly because PM_{2.5} absorbs and scatters sunlight. We also note that SO₂ and CO were negatively correlated with atmospheric visibility, and O₃ was positively correlated with atmospheric visibility, largely reflecting the relationships between these pollutants and PM_{2.5}. This is supported by Supplementary Appendix Table S11, which shows the relationship between air pollution exposure and jaundice performed by stratified analysis of visibility.

We now clarify that in this study population, the newborns stayed in the wards all the time before they were discharged from the hospital, and thus they were not affected by sunlight.

Table R9. Correlations between different air pollution exposures as well as between each air pollutant and atmospheric visibility

	SO ₂	O ₃	CO	Visibility
PM _{2.5}	0.542	-0.250	0.849	-0.667
SO ₂		-0.366	0.635	-0.376
O ₃			-0.398	0.074
CO				-0.490

Question: *A major contribution of this study is elucidating the impact of breathed air pollution on Jaundice risk and the analyses should therefore carefully control for sunlight exposure (impacted by season and atmospheric visibility) in the analysis of breathed air pollution risk. Stratified analysis by TOA and atmospheric visibility would help here.*

Our reply: We now clarify that in this study population, the newborns stayed in the wards all the time before they were discharged from the hospital, and thus they were not affected by sunlight. This is supported by Supplementary Appendix Table S10, which shows the relationship between air pollution exposure and jaundice performed by stratified analysis of TOA irradiance.

Supplementary Appendix Table S11 shows the relationship between air pollution exposure and jaundice performed by stratified analysis of visibility. Atmospheric visibility was categorized into four levels separated by the 25%, 50% and 75% percentiles of the atmospheric visibility range: below 4.7 km, 4.7-8.2 km, 8.2-15.9 km, and above 15.9 km. Overall, the air pollution-jaundice relationship does not affected by the level of visibility.

Question: *Table 1 does not include any socio-demographic characteristics. Are these available in the health records? The fact that there was such a strong different in hypertension in pregnancy is surprising.*

Our reply: On page 19, lines 436-444 of the original manuscript, we had described the socio-demographic characteristics, i.e. “For each newborn, we collected his/her mother’s information such as age, occupation, educational level, gravidity, ..., and blood pressure. Neonatal characteristics including gender, height, weight, Apgar Score, infant special cases, and the delivery process like fetal distress, umbilical cord, and amniotic fluid.”

Pregnant women with different sociodemographic characteristics may be exposed to different air pollution levels. Supplementary Appendix Table S2 shows that female farmers had the higher risk associated with their babies’ jaundice, compared to their counterparts. In China, farmers usually had a low socioeconomic status and engaged in outdoor work (please the references [4] at the end of this document). Thus, female farmers were more exposed to ambient air pollution than office workers.

The revised main manuscript, Supplementary Material and all required files have been submitted to the journal. We would like to thank you again for your valuable suggestions and comments that have helped us significantly improve the technical context and presentation of our manuscript. We look forward to your hopefully positive decision regarding the publication of our manuscript.

Sincerely,

Liqiang Zhang, Weiwei Liu, Jintai Lin, Kun Hou, Chenghu Zhou, Bo Huang, Xiaohua Tong, William Rhine, Jinfeng Wang, Ying Jiao, Ziwei Wang, Ruijing Ni, Mengyao Liu, Liang Zhang, Ziyue Wang, Yuebin Wang, Yanhong Wang

References

[1] Newborns Section of Pediatrics Branch of Chinese Medical Association.

Proceedings of the National Symposium on Neonatal Jaundice and Infection.
Chinese Journal of Pediatrics, 39 (3), 184 – 187 (2001).

- [2] The Editorial Board of Chinese Journal of Pediatrics, et al. Experts consensus on principles for diagnosis and treatment of neonatal jaundice. Chinese Journal of Pediatrics, 48(9), 685-686 (2010).
- [3] Liang, X., et al. PM_{2.5} data reliability, consistency, and air quality assessment in five Chinese cities. J. Geophys. Res. Atmos., 121, 220-10, 236 (2016).
- [4] Wang, Y., et al. Association of Long-term Exposure to Airborne Particulate Matter of 1 μm or Less with Preterm Birth in China. JAMA Pediatr., 8(8), e174872(2018).

REVIEWERS' COMMENTS:

Reviewer #1 (Remarks to the Author):

The revised version has addressed earlier concerns with the manuscript.

Reviewer #2 (Remarks to the Author):

The authors fully addressed my major concerns regarding confounding of the air pollution associations by sunlight exposure. This analysis provides robust new information on the potential association between air pollution exposure and jaundice risk. Below are minor comments.

- 1.) While newborns were in the ward (and therefore had limited exposure to outdoor sunlight), mothers were exposed to different levels of sunlight during pregnancy, which could confound the air pollution effects (especially for maternal exposure) This should be mentioned in the discussion.
- 2.) To clarify the stratified analyses by TOA and visibility I would add an overall linear model for the entire exposure range, in addition to the exposure intervals presented.
- 3.) The stratified analyses demonstrate no pattern of air pollution effects by TOA or visibility (or even higher PM2.5 results at higher visibility), which is strong evidence that this result is not driven by confounding. A sentence could be added to the abstract to highlight this point.
- 4.) The association between exposure time and jaundice risk should be clarified in the results section, in terms of how air pollution levels are incorporated. This is provided in the methods section but there is no detail in the results section which makes interpretation difficult.
- 5.) Line 285 – thus they were not exposed to outdoor sunlight.

Response to Reviewers' Comments for Manuscript

NCOMMS-18-30209A

Risk of neonatal jaundice associated with air pollution exposure

Dear Reviewers,

We would like to thank you for your valuable comments that helped us revise and improve the presentation and technical context of our manuscript. In our individual replies, we have addressed the comments and suggestions of Reviewer 2. The revised manuscript and Supplementary Material now includes appropriate modifications that have been made possible from your very thoughtful comments.

The modifications made in the revised manuscript are highlighted using a yellow background.

Response to Reviewer 2

Question: The authors fully addressed my major concerns regarding confounding of the air pollution associations by sunlight exposure. This analysis provides robust new information on the potential association between air pollution exposure and jaundice risk.

Our reply: Thanks very much for the positive comments. We have addressed all of your comments, and revised the main manuscript and Supplementary Material.

Question: While newborns were in the ward (and therefore had limited exposure to outdoor sunlight), mothers were exposed to different levels of sunlight during pregnancy, which could confound the air pollution effects (especially for maternal exposure). This should be mentioned in the discussion.

Our reply: We have mentioned it in the Concluding Remarks section of the revised manuscript.

Question: To clarify the stratified analyses by TOA and visibility, I would add an overall linear model for the entire exposure range, in addition to the exposure intervals presented.

Our reply: We have added the following linear model for the entire exposure range.

$$Y=\varepsilon+\alpha PM_{2.5}+\beta SO_2+\gamma CO \quad (1)$$

where Y is the bilirubin level. ε is the intercept. α , β and γ are the parameters of air pollutants $PM_{2.5}$, SO_2 and CO , respectively.

Question: The stratified analyses demonstrate no pattern of air pollution effects by TOA or visibility (or even higher $PM_{2.5}$ results at higher visibility), which is strong evidence that this result is not driven by confounding. A sentence could be added to the abstract to highlight this point.

Our reply: We have added the sentence “The jaundice-pollution relationship is not affected by top-of-atmosphere solar irradiance and atmospheric visibility” to the abstract to highlight this point.

Question: The association between exposure time and jaundice risk should be clarified in the results section, in terms of how air pollution levels are incorporated. This is provided in the methods section but there is no detail in the results section which makes interpretation difficult.

Our reply: In the section “Associating neonatal exposure time and jaundice risk”, we have clarified how the average pollution level was controlled as confounders: “After the influences on the peak bilirubin levels of average pollutant concentration and the interaction with the exposure time was controlled (see Methods), the relationship between the exposure time and the peak bilirubin level of each newborn was determined”.

Question: Line 285 – thus they were not exposed to outdoor sunlight.

Our reply: We have modified the sentence in the revised manuscript.

We would like to thank you again for your valuable suggestions and comments.

Sincerely,

Liqiang Zhang, Weiwei Liu, Kun Hou, Jintai Lin, Changqing Song, Chenghu Zhou, Bo Huang, Xiaohua Tong, Jinfeng Wang, William Rhine, Ying Jiao, Ziwei Wang, Ruijing Ni, Mengyao Liu, Liang Zhang, Ziyi Wang, Yuebin Wang, Xingang Li, Yanhong Wang